# Gingival Cyst of the Adult: A Case Description with a Relevant Literature Analysis

**DOI:** 10.3390/reports7030051

**Published:** 2024-06-24

**Authors:** Marta Forte, Antonio d’Amati, Alfonso Manfuso, Massimiliano Vittoli, Giorgia Girone, Eliano Cascardi, Saverio Capodiferro

**Affiliations:** 1Department of Interdisciplinary Medicine, University of Bari ‘Aldo Moro’, 70110 Bari, Italy; alfonso.manfuso@policlinico.ba.it; 2Department of Precision and Regenerative Medicine and Ionian Area (DiMePRe-J), University of Bari ‘Aldo Moro’, 70100 Bari, Italy; antonio.damati@uniba.it (A.d.); eliano.cascardi@policlinico.ba.it (E.C.); 3Private Practice, 70110 Bari, Italy; m.vittoli@virgilio.it (M.V.); giorgiagirone@hotmail.com (G.G.)

**Keywords:** odontogenic cyst, gingival cyst, oral cavity

## Abstract

Gingival cysts of the adult are rare and benign odontogenic lesions of the oral cavity, accounting for almost 0.3% of all odontogenic cysts. Their differential diagnosis is still challenging for surgeons as it includes other gingival inflammatory or non-inflammatory lesions and peripheral odontogenic tumors. The aim of this paper is to report a new case occurring in an adult, analyzing the clinical, radiographic, and histopathological features as guidelines for a differential diagnosis. We report a 49-year-old patient complaining of a small, pigmented lesion localized on the attached gingiva with no history of trauma, which was surgically excised and histologically diagnosed as a gingival cyst. A differential diagnosis may be challenging for clinicians it includes a wide spectrum of inflammatory and non-inflammatory lesions, so a correct diagnostical–therapeutical approach is needed to avoid possible overtreatment and minimize the recurrence rate.

Gingival cyst of the adult (GCA) is a rare and benign odontogenic entity which accounts for about 0.3% of all odontogenic cysts; it arises from the dental lamina, but its etiology remains unclear; some hypotheses suggest a possible origin in the heterotopic glandular or enamel residues, dental lamina, or periodontal ligament [1,2]. A possible association with trauma has been reported as well [1,2,3]. The most common localization described in the literature is the vestibular attached gingiva in the mandibular canine and first premolar areas; the cysts mainly occur in women during the fifth and sixth decades of life and generally grow as a slow and painless swelling, usually solitary and small (measuring about 3–4 mm) nodules or vesicles varying in color from flesh-colored to bluish due to the presence of cystic fluid; multiple occurrences are poorly described [3,4,5,6,7]. GCAs usually lack bone involvement on radiograms, but bone resorption may be observable in relation to the cystic fluid pressure [8]. Upon histopathological examination, a GCA is characterized by a subepithelial connective tissue wall covered by a thin, squamous or cuboidal epithelium, where, in some points, glycogen-rich clear cells might be found [9,10]. The differential diagnosis includes gingival inflammatory or non-inflammatory lesions such as a fibroma, epulis or lateral periodontal cysts, peripheral odontogenic tumors (e.g., peripheral ameloblastoma or peripheral odontogenic keratocysts), and vascular lesions [2,3,4,5,6,7,8,9,10,11,12]. The gold standard treatment for GCAs remains surgical excision, which is associated with a minimally invasive mucosal approach and very low recurrence rate [4,5,6,7,8,9,10,11,12,13].

We describe a case of a gingival cyst occurring in an adult patient and analyze the clinical, radiographic, and histopathological features as guidelines for a differential diagnosis; an analysis of the related relevant literature is also discussed.

A 49-year-old female patient reporting a small, pigmented lesion on the upper adherent gingiva between the central and lateral incisors in the II quadrant was brought to our attention. The patient reported its asymptomatic occurrence about 6 months earlier and reported no history of trauma. Intraoral examinations revealed a single small nodular lesion of the attached gingiva (at the interdental papilla), measuring approximately 3 mm and blueish in color (Figure 1). A periapical radiograph showed no bone alterations related to the lesion (Figure 2). Surgical excision by blade and with an incision in the wide and deep margins was performed with direct closure by stitches (Vicryl 5/0). A microscopic evaluation of the sample revealed a mucosal fragment with a cyst in the subepithelial connective tissue covered by a cubic, stratified epithelium measuring 0.66 mm in size, leading to a final diagnosis of gingival cyst (Figure 3; Figure 4). No post-operative complications were observed, and no signs of recurrence were detectable during follow-up (Figure 5).

GCAs are classified as an uncommon odontogenic lesions which occur mostly in the vestibular attached gingiva and mainly in the mandibular canine and first premolar area; because of their rarity and uncommon clinical appearance, GCAs always represent a challenge in differential diagnosis as including inflammatory lesions (e.g., gingivitis, fibrous hyperplasia, epulis fissuratum, peripheral giant cell granulomas, and pyogenic granulomas) [4,5,11,12] and gingival tumors (e.g., peripheral fibromas or peripheral ossifying fibromas), as well as developmental lesions (e.g., lymphangioma, vascular malformation) and peripheral odontogenic lesions (e.g., peripheral odontogenic fibromas and peripheral ameloblastomas) [11].

The most challenging differential diagnosis involves lateral periodontal cysts (LPCs) as both originate from the odontogenic epithelial rests of the dental lamina. LPCs and GCAs have similar histopathological features as well and, for this reason, clinical and radiographic findings are necessary to define the diagnosis. Nikitakis et al. described in their study a radiographic image of LPC associated with a well-circumcised intraosseous radiolucency at the level of the dental mid-root or lateral root surface. [14] GCAs are rarely evident on radiograms; however, when the cyst pressure determines bone involvement, they may appear as a crestal bone translucency, making it impossible to differentiate GCAs from LPCs [8].

A differential diagnosis may also include peripheral manifestations of odontogenic lesions, mainly in the early stage, such as peripheral keratocystic odontogenic tumors, peripheral ameloblastomas, peripheral ossifying fibromas, and peripheral giant cell granulomas [11]. GCAs usually have a low recurrence incidence after complete excision, and their clinical presentation is described as slow-growing, painless swelling. Peripheral manifestations of odontogenic tumors have more aggressive clinical behavior and are possibly associated with paresthesia, pain and facial swelling, and more evident bone involvement [15,16,17].

GCAs have been described in the literature for over 70 years, though, as pointed out by Frigerio et al. in their study, they have been reported as odontogenic cysts or as a subcategory without a detailed description [3,18]. Ritchey and Orban were the first to study a GCA as an independent lesion, though it was considered a histological entity without clinical interest [19]. GCAs were classified as distinct lesions in a study by Bhaskar and Laskin, who accurately described three occurrences in women (40, 38, and 43 years old), all of which appeared as nonpainful, circumscribed masses on the attached gingiva (mandibular canine and premolar), also suggesting a traumatic origin and differentially diagnosing GCAs from periapical lesions, mental foramina, and central jaw cysts [20]. Therefore, in their study, Frigerio et al. increased the number of details, reporting 15 cases of GCAs, considering age, gender, localization, size, radiographic evidence, and bone involvement for each case. The limit of their study concerns the ages of the patients, which ranged only from 68 to 82 years, reducing the review of the literature on GCAs; among the reported cases, the majority occurred in the mandibular region, precisely on the attached gingiva in the lateral incisor–canine–premolar region and mainly in women, with no history of trauma, and the differential diagnosis included gingival swellings as fibromas or retained foreign bodies [3].

A wide literature review of the literature was conducted in 2009 by Kelsey at al., who collected a total of 155 cases described in detail (by gender, localization, age, race, radiographic evidence, and osseous involvement); in their study, 80% of lesions were localized in the mandible, mainly in the canine–premolar area, and they were mostly diagnosed in female patients of about 49 years of age, while radiographic evidence (as a circular or elliptical radiolucent area with radiopaque border) was detectable in less than 25% of cases [13].

In the current study, we report a GCA of the vestibular attached gingiva between elements #21 and #22; it was painless, slow-growing, and without bone involvement on radiograph, and it was treated by surgical excision. Although this study is limited to only one case report and a classical wide-margin surgery was performed, the data from the literature discussed herein should lead readers to consider the variable clinical and radiological appearance of GCAs which may cause difficulties with differential diagnosis, finally resulting in unnecessary surgical overtreatment.

## Figures and Tables

**Figure 1 reports-07-00051-f001:**
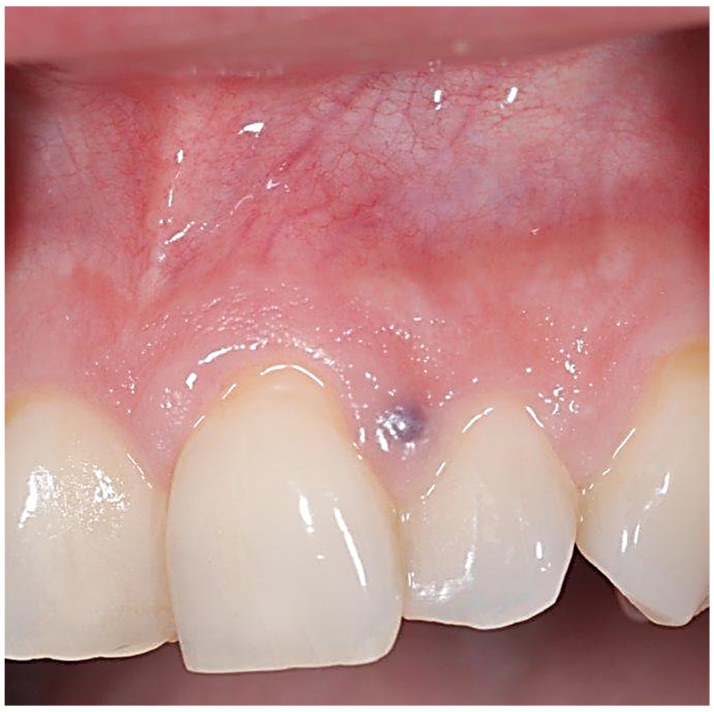
A bluish, solitary lesion on the interdental papilla between teeth #21 and #22.

**Figure 2 reports-07-00051-f002:**
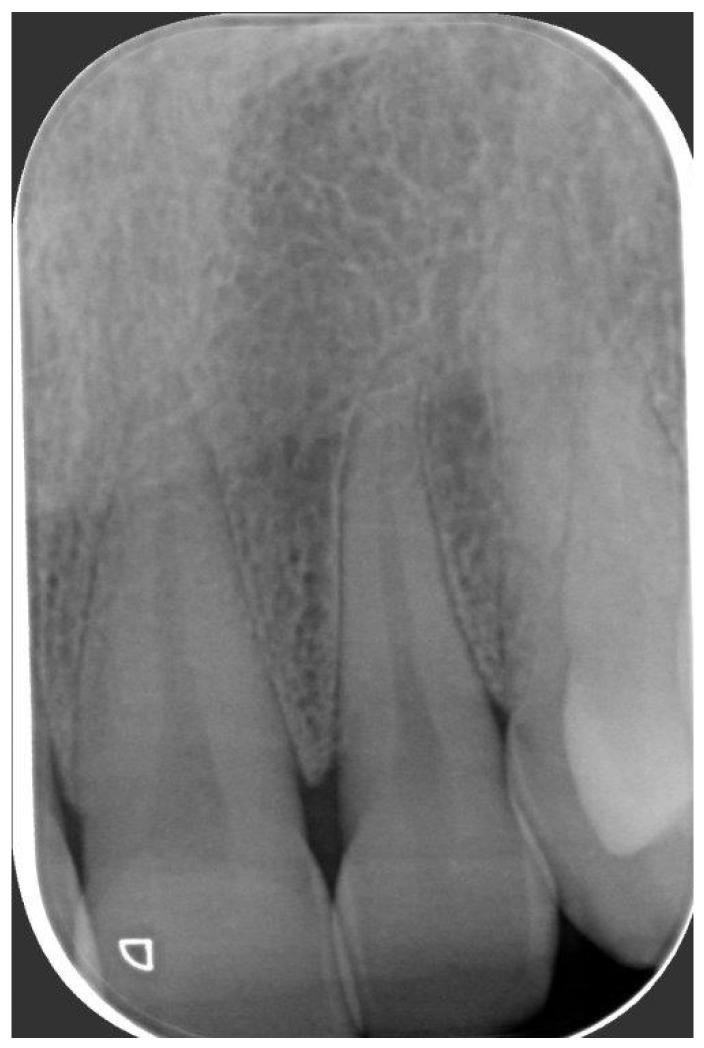
Periapical radiogram showing no alveolar bone involvement.

**Figure 3 reports-07-00051-f003:**
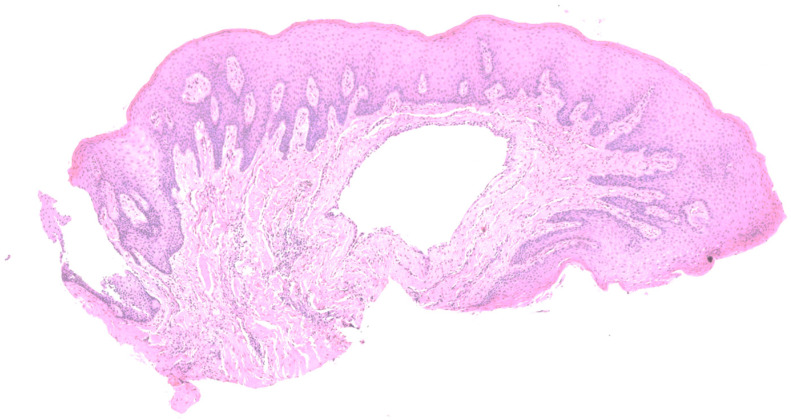
Low-power magnification of the sample showing a cystic lesion located in the lamina propria (H&E, X2).

**Figure 4 reports-07-00051-f004:**
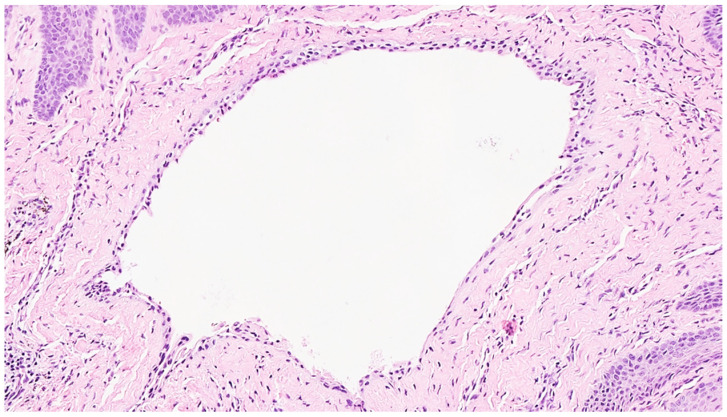
High-power magnification showing a very thin squamous epithelium, leading to the diagnosis of gingival cyst (H&E, X10).

**Figure 5 reports-07-00051-f005:**
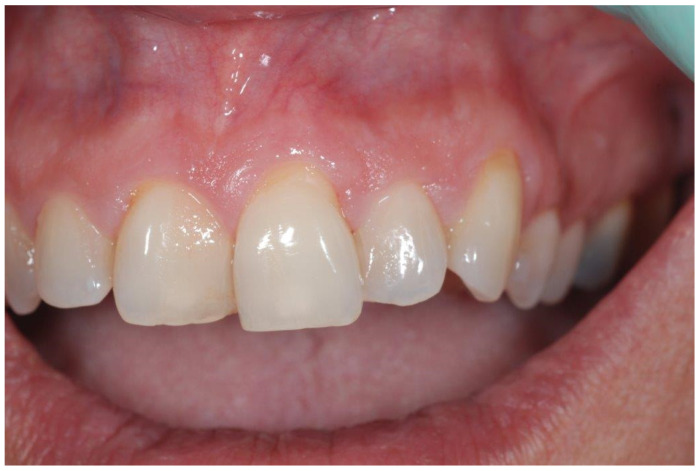
Complete healing after 30 days without sign of recurrence.

## Data Availability

The data presented in this study are available on request from the corresponding author due to privacy restrictions.

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
