# Peer review of "Gingival Cyst of the Adult: A Case Description with a Relevant Literature Analysis"

_reports, 2024, doi:10.3390/reports7030051_

Round 1
Reviewer 1 Report
Comments and Suggestions for Authors
The manuscript could be interesting for the journal readers. However, I have some important concerns about this kind of report.
1. Abstract: Please prepare the abstract according to the requirements of the journal (without subheadings).
2. Keywords: Please review if these words are MeSH terms.
3. Background: Please explain with clarity the scientific rationale for this study. It is important to define the study justification in terms of what is the main contribution to the literature.
4. Methods- Results- Discussion:
Please prepare the manuscript according to the CARE guidelines available in: https://www.equator-network.org/
Please mention the limitations of this report since this kind of study has the low scientific evidence.
Please mention implications for research in dentistry.
Author Response
- Abstract: Please prepare the abstract according to the requirements of the journal (without subheadings).
DONE
- Keywords: Please review if these words are MeSH terms.
DONE
- Background: Please explain with clarity the scientific rationale for this study. It is important to define the study justification in terms of what is the main contribution to the literature.
DONE
- Methods- Results- Discussion:
Please prepare the manuscript according to the CARE guidelines available in: https://www.equator-network.org/
DONE
Please mention the limitations of this report since this kind of study has the low scientific evidence.
DONE
Please mention implications for research in dentistry.
DONE
Reviewer 2 Report
Comments and Suggestions for Authors
This case report has nothing new to learn for the reader.
Author Response
This case report has nothing new to learn for the reader.
The report is about a rare occurence of odontogenic lesion; we decided to report it as well documented clinically and histologically too, and also because a discussion about the cases reported in literature has been included, mainly pointing the attention on the differential diagnosis and the differences in radiological appearance.
Reviewer 3 Report
Comments and Suggestions for Authors
I thank the authors for choosing to share their case of adult gingival cyst with the scientific community.
The manuscript is clear and easy to read, however I would suggest improving the English sentence constructions that seem to be a translation of another spoken language.
With regard to the presentation of the case, I would suggest expanding the description, going into more detail, both on the patient's characteristics, e.g. whether she was taking any medication or had any pathology, and on other aspects (e.g. how the surgery was performed and with what instruments).
Finally, I believe that, considering the authors' stated aim ’We describe a case of gingival cyst occurring in an adult patient to analyse the
clinical, radiographic and histopathological features as guidelines for the differential diagnosis; an analysis of the related relevant literature has been discussed too', a more systematically elaborated discussion is necessary to analyse the differential diagnosis from a clinical, radiographic and histopathological point of view.
Currently from 94 to 101 there is the clinical differential diagnosis with some lesions, and from 85 to 93 histological and a hint of radiographic but with another type of lesion.
The manuscript is well structured, some improvements could make it even more comprehensive.
Line 102: check the abbreviation GCA.
Comments on the Quality of English LanguageThe manuscript is clear and easy to read, however I would suggest improving the English sentence constructions that seem to be a translation of another spoken language.
Author Response
The manuscript is clear and easy to read, however I would suggest improving the English sentence constructions that seem to be a translation of another spoken language.
DONE TROGHOUT THE PAPER
With regard to the presentation of the case, I would suggest expanding the description, going into more detail, both on the patient's characteristics, e.g. whether she was taking any medication or had any pathology, and on other aspects (e.g. how the surgery was performed and with what instruments).
DONE
Finally, I believe that, considering the authors' stated aim ’We describe a case of gingival cyst occurring in an adult patient to analyse the
clinical, radiographic and histopathological features as guidelines for the differential diagnosis; an analysis of the related relevant literature has been discussed too', a more systematically elaborated discussion is necessary to analyse the differential diagnosis from a clinical, radiographic and histopathological point of view.
WE WELL UNDERSTAND THE SUGGESTION OF REVIEWR 3, BUT WE DECIDE TO DISCUSS IN THE REPORT ONLY THE MAIN PAPERS ABOUT THE TOPIC, CHOOSING THEM BETWEEN THE PAPERS WITH LARGE NUMBER OF CASES AS TO REPORT MORE DATAILES ABOUT THE CLINNICAL AND RADIOLOGICAL APPEARANCE OF GC. FOR THE SAME REASON THE PAPER WAS SUBMITTED AS CASE REPORT AND NOT AS REVIEW OF THE LITARATURE
Currently from 94 to 101 there is the clinical differential diagnosis with some lesions, and from 85 to 93 histological and a hint of radiographic but with another type of lesion.
IN THIS SECTION WE TRY TO COLLECT ALL DATA ABOUT DIFFERENTIAL DIAGNOSIS EMERGIN FROM THE STUDY PUBLISHED IN LITERATURE REPORTING ON LARGE NUMBER OF GC
The manuscript is well structured, some improvements could make it even more comprehensive.
DONE
Line 102: check the abbreviation GCA.
DONE
Round 2
Reviewer 1 Report
Comments and Suggestions for Authors
The manuscript can be accepted
Reviewer 2 Report
Comments and Suggestions for Authors
I do not think so; it is helpful for the reader. Kindly upload the post-surgical photo (after the surgical removal of the cyst) and the cyst photo, and the suturing photo. It may be rare, but it is readily available on the internet. you can find the following article
Mota ME, Oliveira DMA, Medeiros YL, Moreira MS, Lopes RN, Alves FA, Louredo BVR, Vargas PA, Prado JD. Gingival cyst of the adult. Autops Case Rep. 2023 Nov 13;13:e2023454. doi: 10.4322/acr.2023.454. PMID: 38034522; PMCID: PMC10688211.
Frigerio M, Al Eid R, Lombardi T. A Rare Odontogenic Cyst: Gingival Cyst of the Adult. A Case Report and Review of the Literature in Elderly Patients. Case Rep Dent. 2023 Jul 26;2023:4827611. doi: 10.1155/2023/4827611. PMID: 37546573; PMCID: PMC10397494.